# Preclinical Ultrasonography in Rodent Models of Neuromuscular Disorders: The State of the Art for Diagnostic and Therapeutic Applications

**DOI:** 10.3390/ijms24054976

**Published:** 2023-03-04

**Authors:** Antonietta Mele, Paola Mantuano, Brigida Boccanegra, Elena Conte, Antonella Liantonio, Annamaria De Luca

**Affiliations:** Section of Pharmacology, Department of Pharmacy-Drug Sciences, University of Bari “Aldo Moro”, 70125 Bari, Italy

**Keywords:** ultrasonography, skeletal muscle, neuromuscular disorders, translational research

## Abstract

Ultrasonography is a safe, non-invasive imaging technique used in several fields of medicine, offering the possibility to longitudinally monitor disease progression and treatment efficacy over time. This is particularly useful when a close follow-up is required, or in patients with pacemakers (not suitable for magnetic resonance imaging). By virtue of these advantages, ultrasonography is commonly used to detect multiple skeletal muscle structural and functional parameters in sports medicine, as well as in neuromuscular disorders, e.g., myotonic dystrophy and Duchenne muscular dystrophy (DMD). The recent development of high-resolution ultrasound devices allowed the use of this technique in preclinical settings, particularly for echocardiographic assessments that make use of specific guidelines, currently lacking for skeletal muscle measurements. In this review, we describe the state of the art for ultrasound skeletal muscle applications in preclinical studies conducted in small rodents, aiming to provide the scientific community with necessary information to support an independent validation of these procedures for the achievement of standard protocols and reference values useful in translational research on neuromuscular disorders.

## 1. Introduction

Ultrasonography is a non-invasive and patient-friendly diagnostic imaging technique that uses high-frequency sound (ultrasound) waves to produce visual images of organs and tissues. Thanks to its numerous advantages, namely use of non-ionizing radiation, real-time display, portability and relatively low costs, ultrasonography has been employed in clinical practice since 1950 in several fields of medicine, such as cardiology, gynecology, gastroenterology, and oncology, as well as to evaluate property changes of nerves and tendons, skin fibrosis and/or connective tissue disease progression [1,2,3]. In 1980, Heckmatt et al. published a paper in which ultrasonography was used for the first time to evaluate the structural changes in skeletal muscle, demonstrating clinically relevant differences in the thigh structure between children with muscular dystrophy and healthy controls [4]. From then on, ultrasonography has been successfully and extensively used for studying skeletal muscle in clinical practice. Indeed, ultrasonography evaluations are very useful for the diagnosis of several pathophysiological conditions involving skeletal muscle, as well as for monitoring muscle damage and changes occurring in the progression of chronic neuromuscular disorders and/or related to physical activity. This, consequently, also allows monitoring of the efficiency of therapeutic interventions. In spite of the extensive clinical use of ultrasonography for skeletal muscle evaluations, there is a still limited availability of standardized protocols to be applied in preclinical studies on rodent animal models of physiopathological conditions primarily or secondarily involving skeletal muscle. Hence, to contribute to filling this gap, this review represents the first endeavor to recapitulate the most recent advancements regarding ultrasonography skeletal muscle applications and image acquisition modalities in preclinical studies conducted in small rodent experimental models. A final aim is to deliver up-to-date indications to promote the rigorous application of this technique in preclinical settings and the future development of standard protocols and reference values useful to improve translational research on neuromuscular disorders. 

### 1.1. Clinical Application of Ultrasonography for Skeletal Muscle in Health and Disease

To better understand the translational potential of ultrasonography for the study of skeletal muscle, we believe it is useful to briefly summarize its main clinical applications in this field. To date, in hundreds of studies, ultrasonography has been used to examine skeletal muscle structural and morphological parameters in different human skeletal muscle types. Skeletal muscle ultrasonography acquisitions allow measuring specific parameters such as pennation angle (PA), fiber length (FL), muscle thickness (MT), cross-sectional area (CSA), volume, fat and/or fibrotic infiltrations. All of these features represent an indirect measure of muscle function and morphology in different physiopathological conditions. For instance, the hypertrophy observed in the triceps brachii muscle of bodybuilders is well correlated with the increase in ultrasonography-measured PA; this is in line with the functional significance of the pennation, representing a strategy to pack in parallel a larger number of contractile elements along the tendon [5]. Accordingly, gastrocnemius muscles of patients affected by unilateral disuse-induced muscle atrophy are characterized by a decrease in CSA of the injured leg, as well as by a decrease in PA and FL, suggesting that disuse-induced atrophy involves a loss of both parallel and in-series sarcomeres [6,7]. Furthermore, the MT, measured by ultrasonography, also represents an index of muscle atrophy [8], as demonstrated by the good correlation observed between this parameter and muscle strength and electrophysiology findings [9]. Moreover, in mammalian skeletal muscle, these parameters were shown to influence functional properties, such as length–tension relationship, tension per unit weight and force–velocity (F–V) properties [10]. A recent review, which included 125 original studies on the assessment of aging-related muscle loss, concluded that ultrasonography is a valid and reliable imaging technique for the assessment of muscle wasting. In particular, the most widely used MT parameter also turned out the best to describe this phenomenon, showing a good correlation with other standard measurements of muscle loss using dual energy X-ray absorptiometry (DEXA), magnetic resonance imaging (MRI) and computed tomography (CT). However, the authors also concluded that three-dimensional ultrasound measurements of muscle volume could provide better results to investigate muscle loss [11]. 

In parallel with the evaluation of muscular trophism, the ultrasound technique allows one to perform quantitative evaluations of skeletal muscle echogenicity. As previously described, a healthy muscle appears predominantly black with a low echo intensity. In various physiopathological conditions, the presence of adipose or fibrotic tissue increases the amount of ultrasound beam reflections, giving back brighter, more echogenic images. For example, in skeletal muscles from patients affected by DMD, the absence of dystrophin leads to an increase in susceptibility to sarcolemmal damage, resulting in repeated cycles of degeneration/regeneration and inflammation that induce a progressive replacement of muscle by fibrous and fat tissue [12,13,14,15]. In patients already diagnosed with DMD by phenotype and genetic testing, ultrasonography represents a resourceful tool to monitor the disease severity and progression [16]. In this regard, quantitative ultrasonography evaluations of skeletal muscle echogenicity in young DMD patients have been correlated with clinical parameters used for the assessment of disease progression such as muscle strength, ambulatory status, and motor ability [17].

### 1.2. Skeletal Muscle Ultrasonography in Preclinical Studies

Thanks to the development of high-resolution ultrasound devices, it is currently possible to perform acquisitions of rodent skeletal muscles. As shown in Figure 1, which shows a sample image of rat’s hind limb muscles, ultrasonography allows one to discriminate different muscles thoroughly. In fact, each muscle, due to its predominantly aqueous composition of larger myofibers, appears substantially as a hypoechoic structure and, therefore, black. However, the boundaries of each muscle appear as hyperechoic white structures due to highly reflective structures such as the epimysium, a connective tissue membrane surrounding the whole muscle. Furthermore, inside each muscle, it is also possible to visualize muscle bundles and their direction, since these latter are also surrounded by a highly reflective hyperechoic connective membrane, the perimysium (Figure 1).

Despite its broad application for skeletal muscle evaluations in clinical settings, ultrasonography lacks validated references for preclinical applications. Over the past ten years, the growing demand for non-invasive imaging techniques in preclinical settings has driven the technological evolution of the ultrasound tools, resulting in high-frequency micro-ultrasound systems able to resolve tissues from living small animals such as rodents [18]. Much progress has been made mostly for non-invasive echocardiographic assessment of cardiac function in different animal models [19], leading to the development of standard protocols and guidelines, such as in the dystrophic *mdx* mouse model of DMD (TREAT-NMD SOP(ID) Number: DMD_M.2.2.003; https://treat-nmd.org/resources-support/research-overview/preclinical-research/experimental-protocols-for-dmd-animal-models; access date 4 June 2015. However, to date, only a few reports have investigated the functional and morphological alterations of rodent skeletal muscles by using ultrasonography. This hamper obtaining a proper standardization of ultrasonography preclinical readouts to evaluate either pathology progression or potential efficacy of treatments in animal models of neuromuscular disorders, which may help to improve the translation of preclinical data as well as the application of ultrasonography at the clinical level. 

In addition, ultrasonography may represent a valid in vivo, non-invasive alternative to assess skeletal muscle changes in experimental models of pathophysiological conditions. Indeed, with respect to the conventional ex vivo methods for the assessment of muscle features, ultrasonography allows one to perform longitudinal studies, thus decreasing the number of animals needed to obtain statistically significant results. 

In this narrative review, we firstly revised the common preparative details to perform an ultrasonography evaluation of skeletal muscle (anesthesia and animal preparation), as well as the choice of the most suitable probe. Then, we focused on the methodological modalities used to study skeletal muscle features in rodents, based on the studies available on the main bibliographic databases, selected by using specific criteria described later in the text. These studies used ultrasonography to investigate structural and morphological parameters in rodent models (mouse and rat) of conditions/diseases in which skeletal muscle is primarily or secondarily affected, with particular attention to hind limb muscles and diaphragm. The final aim was to provide the scientific community with a single work encompassing the necessary information to support further preclinical studies using ultrasonography on skeletal muscle, promoting an independent validation of these procedures. 

## 2. Methods

In this narrative review, we conducted a literature search to find studies using ultrasound as the main method to investigate structural and morphological skeletal muscle parameters in rodents (rats and mice) in several pathophysiological conditions. To do this, we used Scopus and PubMed bibliographic database searching and crossing with the “AND” Boolean operator and the following terms: “skeletal muscle”, “rat”, “mouse”, “rodent”, “ultrasonography”, “muscle volume”, “fiber length”, “muscle thickness”, “pennation angle”, “cross-sectional area”, “diaphragm movement”, and limited our search to original research articles published in English and in peer-reviewed Journals. For the review, we selected only the studies where the protocols were carefully reported, also providing details about animal position, probe choice and parameter analysis. Articles were excluded if they did not meet the aforementioned inclusion criteria. A total of 12 articles were finally selected.

## 3. Results

### 3.1. Preliminary Steps for Skeletal Muscle Ultrasonography Evaluations

#### 3.1.1. Anesthesia

For skeletal muscle evaluations, rats and mice are usually anesthetized via inhalation with isoflurane [20]. For the induction phase, we recommend ~3–3.5% isoflurane and 1.5 O_2_ followed by a maintenance phase of ~1.5–2% isoflurane and 1.5 O_2_. To work at low anesthesia levels is of utmost importance, especially during the evaluation of diaphragm function. Indeed, anesthesia significantly influences the respiratory and heart rate, important physiological parameters, while monitoring diaphragm function. By using isoflurane at the described concentrations, a respiratory rate (breaths/min) of 100–120 and a heart rate (beats/min) of 300–400 is expected [21].

#### 3.1.2. Animal Preparation

Before starting ultrasound acquisitions, it is necessary to carefully shave the area to be scanned to avoid interferences, by using a depilatory cream. Furthermore, eye lubricant must be placed on each eye to prevent drying of the area, and a small amount of electrocardiographic (ECG) gel on the platform copper leads to allow ECG and respiratory recording. 

Body temperature must be constantly monitored during the imaging session using a rectal probe and maintained within its physiological range (36–37.5 °C) via the heating platform. An ultrasound gel needs to be added between the animal skin and the probe. During acquisitions, minimal pressure is applied to the muscle to minimize image distortion. By the end of the ultrasonography session, animals usually recover from anesthesia in 10–15 min [21].

#### 3.1.3. Probe Choice

The choice of the most suitable probe frequency and the correct positioning of the animal for specific imaging sessions are two key elements to be considered for the standardization of ultrasonography procedures to ensure repeatable data. A limit in reproducibility is indeed often represented by the fact that these details are not always sufficiently described in the papers. However, it must be underlined that the probe frequency influences the resolution of the ultrasonographic image, as well as the penetration power. Higher frequencies are not as penetrating as lower ones, but they ensure a high resolution. For example, a 40 MHz probe is characterized by a high resolution, allowing one to visualize structural characteristics of rat’s skeletal muscle, such as pennation angle and fascicle lengths, as well as to better distinguish the muscle under investigation to perform muscle volume measurements [22,23]. However, a 40 MHz probe does not allow one to perform the same evaluations on mice, in which using a 21 MHz probe is preferrable to obtain the whole hind limb in one image. This strategy made possible a three-dimensional acquisition of all hind limb also in power Doppler mode to evaluate hind limb volume and percentage of vascularization [24]. A detailed description of the most used Vevo 2100 imaging system probes specifically formulated for preclinical settings is shown in Table 1. Here, “Primary Application” refers to the applications reported by the company producing the ultrasonographic apparatus and probes (VisualSonics, Toronto, ON, Canada), while “Skeletal Muscle Applications” refers to the applications on skeletal muscle described in this review. Furthermore, new probes working across a wide range of frequencies have recently been developed for preclinical ultrasonography studies, but none have yet been used on skeletal muscle.

As discussed later, most skeletal muscle acquisitions take place in B-dimensional mode (B-mode), which represents the mode most commonly used in ultrasound studies. A B-mode acquisition displays a two-dimensional image in which the organs or tissues appear in various shades of gray, allowing one to visualize and quantify the anatomical structure parameters. Furthermore, the M-monodimensional mode (M-mode) is a useful ultrasonography mode in which a single scan line is emitted, received, and displayed graphically to observe structures moving at high velocity and evaluate functional parameters as they occur for diaphragm muscle.

### 3.2. Ultrasound Evaluation of Structural and Morphological Parameters of Hind Limb Muscles

In a 2011 study, Peixinho et al. used an ultrasonographic in vivo approach to quantify the changes in two skeletal muscle architectural parameters related to a degeneration–regeneration process subsequent to muscle laceration [22]. Specifically, they evaluated PA (angle between the deep aponeurosis and the line of the fascicle) and MT (distance between the superficial and the deep aponeuroses) from high-resolution B-mode images of rat gastrocnemius (GAS) and soleus (SOL) muscles acquired over time at 0, 7, 21 and 28 days after the laceration (Table 2). Immediately after laceration, the formation of edema and hematoma were responsible for a large hypoechoic area that did not allow the measurement of any structural parameters. On the twenty-first day after the laceration, it was again possible to distinguish the hyperechoic fascicles’ direction and the aponeuroses. This allowed the authors to conduct the PA and MT measurements. PA and MT values of injured GAS and SOL muscles were comparable to the values measured before the laceration. Interestingly, an enhancement of PA and MT was observed in non-injured contralateral GAS and SOL muscles compared with the values obtained from non-injured animals. They explained this as the result of a compensatory hypertrophy occurring on non-injured limb due to the overload condition these muscles were subjected to, by virtue of the laceration of the contralateral limb. The strength of this study lies in its non-invasive ultrasound measurement over time of the architectural parameters influencing the muscle function. The authors stressed that this approach could be particularly important to evaluate the progression of rehabilitation and athletic programs and to obtain mathematical models useful in predicting muscle performance and changes in muscle strength. Despite these important findings, the authors pointed to a limitation in the laceration protocol that did not always induce the same type of injury. Furthermore, they proposed further studies aimed at improving the rat and hind-limb positions more suitable for measuring these parameters [22].

In a following article from the same authors, ultrasonography was used to study the adaptative processes taking place in skeletal muscles in chronic stretching training similar to the protocols used in humans to improve muscle performance. Since skeletal muscle function is strongly influenced by its structure and architecture, the authors proposed to evaluate, by ultrasound, the muscle changes in terms of PA, MT and tendon length (TL) occurring in lateral GAS muscles of rats subjected to chronic stretching. B-mode acquisitions of the lateral GAS muscle were carried out, and PA and MT parameters were measured before and after 6, 12, 18 and 24 h of stretching sessions and compared with those obtained from unstretched control rats (Table 2). They showed a significant and progressive reduction in PA from elongated muscles with respect to non-elongated ones. They concluded that the applied stretching protocol was sufficient to modify the main structural ultrasonographic features of GAS muscle, and this phenomenon may represent an adaptative muscle response to the new condition for optimizing its function [25]. 

The possibility of evaluating muscle injury and subsequent regeneration by using a non-invasive technique assumes greater importance in sports medicine, as well as in monitoring work activities. In this contest, Jimenez-Diaz et al. published a paper in which a model of skeletal muscle injury was generated in rats by locally injecting a 2% solution of the anesthetic mepivacaine hydrochloride into the central portion of GAS muscle of one limb, using the contralateral muscle as the non-injured control. To monitor the skeletal muscle healing process, ultrasound longitudinal and transversal B-mode acquisitions of injured and non-injured GAS muscles were performed at different time points to evaluate several parameters, such as area size of the injured zone, fascial integrity, and injured-area borders. Furthermore, changes in echogenicity were evaluated to detect the possible appearance of anechoic areas due to fluid and/or of hyperechoic areas due to the presence of fibrotic tissue in the injured zone (Table 2). Although ultrasonography by itself was not suitable to distinguish the different phases within the processes of degeneration and regeneration, the combined use of ultrasound imaging and muscle histology allowed the authors to highlight an increase in echogenicity, as typical of the degenerative phase after injury, while the regenerative phase was characterized by a decrease in echogenicity until normal values were reached once healing was achieved. A clear limitation of this study was the use of a clinical apparatus working at lower frequencies [26]. However, these early promising results showed the capability to monitor over time the degenerative/regenerative process in skeletal muscle by ultrasound.

In 2015, Leineweber et al. conducted a pilot study in which they demonstrated that ultrasound could detect the level of muscle damage in a rat contusion injury model, proposing the technique as a valid, non-invasive alternative to functional testing. In this model, the contusion of GAS muscle was obtained by dropping a mass weighting 550 g from a height of 33 cm onto a hemisphere positioned at the level of the muscle. The severity of the injury was evaluated on 360 B-mode images by assigning a numerical score from 0 to 5, with 0 indicating no visible tissue damage, and 5 indicating very severe damage. The scale was constructed by visual observation of damage indicators such as muscle structure, swelling, muscle thickness, muscle boundaries, etc. (Table 2). Ultrasonography score results were correlated with GAS muscle isometric contraction force evaluated by in situ torque test. Interestingly, both functional parameters and ultrasonographic image scores showed a significant injury following contusion and a trend towards recovery over a two-week period [27]. 

Nijhuis THJ et al. proposed an ultrasound technique to measure gastrocnemius thickness and cross-sectional area (diameter) as a valid alternative to the traditional calculation of GAS muscle index (GMI). GMI is considered an indirect measure of muscle force, usually calculated by dividing the weight of GAS muscle after unilateral surgical procedure of sciatic nerve denervation by the weight of the contralateral, not denervated GAS muscle. In this way, it is possible to highlight the effect of denervation by GAS muscle atrophy, while nerve regeneration is usually followed by muscle weight gain. Clearly, this is an invasive procedure requiring the sacrifice of a large number of animals. In their experiments, after surgical procedure, they monitored by ultrasound both denervated and not-denervated rat GAS muscles at regular intervals for 2 months, measuring thickness and diameter. They used an ultrasound system commonly employed for veterinary clinical investigations (Table 2). Their results highlighted a strong correlation between GMI and ultrasonographic gastrocnemius muscle thickness with both devices, while no correlation was found between muscle cross-sectional area and GMI, likely attributable to the intrinsic limitation of correlating a bi-dimensional with a three-dimensional parameter. The authors proposed the GAS muscle thickness as a valid alternative to GMI and claimed that the possibility of correlating GMI to a three-dimensional parameter, such as muscle volume, could overcome the lack of results with cross-sectional area [28]. 

Similarly, the same authors carried out a study on rat tibialis anterior (TA) muscle to validate the ultrasound technique as an alternative to monitor muscle atrophy after denervation. To this end, ultrasound measurements of rat TA cross-sectional area were performed (Table 2) and compared to muscle weight and isometric tetanic force. The tibialis cross-sectional area, as a measure of muscle atrophy, was determined by using Adobe Photoshop CS6 Extended. Data showed a strong correlation between ultrasound tibialis anterior cross-sectional area and muscle weight. A lower, but still significant, correlation was also found between muscle diameter and isometric force. By virtue of these results, the authors proposed ultrasound as a non-invasive, valid new method to determine muscle atrophy after denervation. The authors described a strong correlation between ultrasound tibialis anterior cross-sectional area and muscle weight together with a lower, but still significant, correlation between muscle diameter and isometric force, encouraging them to propose ultrasound as a non-invasive, valid new method to determine muscle atrophy after denervation [29].

More recently, for the first time, ultrasound acquisitions were performed to evaluate rat muscle volume changes in a model of muscle atrophy. In detail, muscle volume was measured on ultrasonographic images obtained from hind limb unloaded (HU) rats, a well-known model of muscular disuse inducing a gradual atrophy of SOL, and to a lesser extent, of GAS muscles [23,34]. After acquisition (Figure 2a), due to the difficulty of acquiring the whole GAS and SOL muscles in their lengths, it was necessary to carry out muscle fragmentation in distal and proximal parts of about the same length, by using a thin strip. The muscle volume calculation was performed by using the truncated cone method, as previously reported for volume calculation in human muscles [35]. This method consists of dividing the muscle into several cross-sectional areas. Then, the total muscle volume was quantified as the sum of all contiguous truncated cones drawn in the distal and proximal parts of the SOL or GAS muscles (Figure 3a). As expected, a progressive and significant reduction in SOL and GAS volume, with a more evident effect on the former, was observed. To validate this new methodology, ultrasound volume, calculated at the end of the suspension period, was correlated to conventional ex vivo readouts, such as muscle volume obtained by muscle weight-to-density ratio and muscle fiber cross-sectional area by cryosections, evaluated on the same animals. They found a good linear correlation both with weight-to-density ratio and fiber cross-sectional area from laminin-stained muscle section for SOL muscle. Due to technical limitation in the excision, a strong linear correlation between the ultrasonographic volume and fiber cross-sectional area from laminin-stained muscle section was found for GAS muscle. Interestingly, they also proposed a new mathematical method for volume calculation. From the observation that in the ultrasonographic window used for the acquisitions, SOL and GAS muscle profiles resembled a sinusoidal function, the muscle volume was also calculated by using a new sinusoidal method in which the proximal and distal SOL and GAS volumes were approximated by the rotation around the tendon-to-tendon axis of a sine function (Table 2; Figure 3b) [23]. This new method showed results comparable to those obtained with the truncated cone method, allowing an easier and faster analysis with a lower margin of error. 

In the light of these results, the authors proposed and validated the ultrasound technique as a new, non-invasive method for preclinical in vivo longitudinal evaluation of rat skeletal muscle atrophy with the advantage of decreasing the number of animals needed to obtain a statistically relevant sample size.

The ultrasound evaluation of muscle volume was later performed on fast-twitch flexor digitorum longus (FDL) muscle in a rat model of cisplatin-induced cachexia (Table 2; Figure 2b). Skeletal muscle loss is considered one of the main characteristics of cancer cachexia and represents a dose-limiting effect of chemotherapy [36]. A multidisciplinary study, involving in vivo and ex vivo approaches, was conducted to gain a deeper understanding of molecular mechanisms contributing to cisplatin-induced cachexia and consequent functional muscle impairment. In addition, the study aimed also to evaluate the potential beneficial effect on muscle cachexia of twice-daily administration for 5 days of hexarelin and JMV2894, two growth hormone secretagogues (GHSs). Importantly, this was the first time that ultrasound was performed before and after a drug treatment in rats [30]. In addition, in this case, FDL muscle was virtually divided into distal and proximal parts approximately of the same lengths, and the muscle volume calculation was performed by using the conventional truncated cone method [35]. Ultrasonographic results highlighted that, in line with the other in vivo and ex vivo readouts (body weight, forelimb force, muscle weight, muscle fiber CSA), cisplatin treatment significantly induced a reduction in FDL muscle volume. 

Interestingly, GHS co-administration was able to prevent cisplatin-induced reduction in FDL muscle volume. In fact, rats treated with hexarelin and JMV2894 showed no significant changes in FDL volume with respect to the pre-treatment condition as well as to vehicle-treated rats. This was in line with muscle weight and cross-sectional area measured ex vivo. This study further validated the use of ultrasound for the non-invasive determination of skeletal muscle structural changes in animal models of muscle wasting, as well as for the evaluation of drug efficacy.

More recently, hind limb ultrasonography was also validated as a reliable and informative technique in preclinical studies conducted in murine models of neuromuscular diseases. A successful application of this approach is represented by several analyses conducted on the *mdx* mouse model, the most widely used animal model for preclinical studies on DMD. DMD is a debilitating genetic disease caused by the loss of dystrophin, a subsarcolemmal protein of the dystrophin–glycoprotein complex that plays a key role in both membrane integrity and mechano-transduction signaling. DMD patients are characterized by progressive skeletal muscle weakness and wasting because of increased susceptibility to contraction injury-inducing myofiber necrosis and the replacement of myofibers by fibrous and fat tissues. Similarly, the absence of dystrophin leads to structural and functional alterations of cardiomyocytes, resulting in a severe cardiomyopathy. However, while the first evidence of skeletal muscle alterations in *mdx* mice occur at an early stage (1–2 months of age) of the pathology progression, cardiac failure represents a late phenomenon, appearing around the 9th month of age [14]. Despite the extensive use of ultrasound to assess the cardiac alterations occurring in *mdx* mice, to date there is no extensive preclinical evidence regarding the application of ultrasonography to assess skeletal muscle features in this model. 

Recently, the structural alterations occurring in 6- and 12-month-old *mdx* hind limb muscles were evaluated by ultrasound using a three-dimensional (3D) approach (Figure 2c). Mouse hind limb volume and percentage of vascularization were obtained by multiple bi-dimensional (B-mode) image acquisitions by translating the ultrasound probe parallel to the long axis of the hind limb (Table 2). At the end of the procedure, 3D images were reconstructed from previously collected multiple frames and visualized with VisualSonics 3D software 1.7.1, allowing calculation of both hind limb total volume and percentage of vascularization. This new approach provided a fast new method to quantify the structural and morphological change occurring in whole *mdx* hind limb skeletal muscle. 

Hind limb ultrasonography results showed a significant increase in total volume in 6-month-old *mdx* mice compared to wild type (WT), in line with the typical skeletal muscle hypertrophy characterizing this mouse model throughout life. In the same study, a group of mice was treated with taurine, a safe amino acid showing a synergistic effect with α-methylprednisolone, the gold standard therapy in DMD patients. However, no effects were exerted by taurine on ultrasonography skeletal muscle volume. This was paralleled by other in vivo results obtained after functional assessment of mouse forelimb grip strength and resistance to exercise, showing that taurine was not able to counteract *mdx* mouse weakness and fatigability. This lack of effect could be possibly related to the chosen stage of the disease, in which skeletal muscle is already severely compromised [37,38].

A similar ultrasonography approach has been used for the skeletal muscle characterization of Kir6.1[V65M] mice, a model of Cantù syndrome, a rare autosomal dominant condition caused by gain-of-function mutations in genes encoding for ATP-sensitive potassium channel subunits [39,40,41]. The 3D evaluation of hind limb muscle volume showed an increase in this parameter related to the presence of fat or fibrotic tissue [31] (Table 2).

### 3.3. Ultrasound Evaluation of Diaphragm Structural and Morphological Parameters

More recently, ultrasonography has been used to longitudinally monitor diaphragm morphology and function in vivo in *mdx* mice. The possibility of monitoring diaphragm muscle by ultrasound over time is particularly interesting, especially considering that the *mdx* mouse model is characterized by diaphragm functional and structural alterations from the early disease stages, showing a progression of dystrophic pathology that closely resembles the one observed in humans. Moreover, ultrasound evaluations of diaphragm morphological and functional alterations have been performed on DMD patients in several clinical studies to monitor disease progression, showing promising results [17,42]. In this context, the validation of the ultrasound technique in the *mdx* mouse model for an in vivo time-dependent evaluation of diaphragm changes could make an important contribution to the acceleration of preclinical translational research. 

Whitehead and colleagues performed diaphragm ultrasound acquisitions for the evaluation of age-dependent changes in movement amplitude as an index of diaphragm muscle function. The experiments were performed on WT and *mdx* mice at different ages (10, 12, 14, 18, and 22 weeks). The amplitude measurements were conducted in M-mode during each inspiration (positive deflection) and calculated as the distance between the baseline and the peak of contraction (Figure 2d and Figure 3c).

The authors showed a significant age-dependent reduction in diaphragm amplitude movement in *mdx* mice with respect to WT mice at every age, in parallel with a strong correlation between this ultrasonography parameter and ex vivo isometric force production evaluated in isolated diaphragm from the same mice, especially at 8 and 18 months of age. Subsequently, the ultrasonography diaphragm amplitude movement was evaluated in 10-week-old *mdx* mice receiving the intravenous administration of AAV-Flag-micro-dystrophin (AAV-µDys), one of the most promising therapeutic strategies for DMD. Diaphragm ultrasound acquisitions were performed before starting the treatment and 2, 4, 6, 8 and 12 weeks after drug injection. Treated *mdx* mice showed a significant increase in diaphragm amplitude as well as the ex vivo specific force value, with respect to age-matched untreated *mdx* mice, with a maximum effect reached after 4 weeks of treatment. This further supports the use of ultrasonographic parameters as valuable preclinical outcome measures to monitor, in vivo and in a non-invasive manner, the efficacy of innovative molecular therapies, alone or in combination, aimed at increasing dystrophin levels [32].

More recently, an independent validation of the ultrasound technique was performed for the longitudinal detection of diaphragm alterations in *mdx* mice, mainly focusing on 3- and 6-month-old animals, since these ages represent the time window in which most pharmacological studies are conducted. In line with previous studies, a significant reduction in diaphragm movement amplitude in both 3-and 6-month-old *mdx* mice compared with age-matched WT was observed. Interestingly, amplitude values correlated well with diaphragm ex vivo specific tetanic force, confirming the early degeneration and functional alteration of this respiratory muscle in dystrophic mice. Furthermore, diaphragm movement amplitude was also well-correlated with the results obtained from a treadmill exhaustion test conducted in the same mice as an index of in vivo muscle fatigue, indicating that the impairment of respiratory function contributes to a progressive increase in muscle fatigue, especially because at this stage of disease, no functional or structural cardiac alterations were observed [32,33] (Table 2). 

In the same study, a first-time evaluation of diaphragm muscle echodensity in dystrophic mice, as an index of pathology progression and contractile tissue loss, was conducted. Diaphragm echodensity was obtained from frames acquired in B-mode from WT and *mdx* mice, and it was measured using ImageJ^®^ software, 1.52a by creating a gray-scale analysis histogram of an outlined diaphragm section of a constant dimension. Moreover, a possible ultrasound attenuation due to abdominal wall was excluded by performing a computational analysis on 36 B-mode images randomly selected from those used for diaphragm echodensity measurements [33]. In 3-month-old *mdx* mice, a trend toward increases in mean pixel echodensity, which becomes significant in 6-month-old *mdx* mice was observed, indicating an age-dependent increment in collagen content in *mdx* diaphragm, described elsewhere, which in turn parallels that observed in patients. Interestingly, in the same mice, an age-dependent increase in muscular pro-fibrotic cytokine transforming growth factor-β1 (TGF-β1) was observed. These results indicated that the ultrasonographic evaluation of diaphragm echodensity is capable of detecting an age-dependent variation of this parameter in *mdx* mice, introducing a novel and highly valuable marker of disease progression in translational research, as it is easily translatable into clinical settings.

## 4. Discussion

Monitoring skeletal muscle properties in small animals such as rodents (mice and rats) by ultrasonography represents more and more a key element in experimental studies aimed at evaluation of the progression of several pathophysiological conditions involving skeletal muscle and the effects of therapies. The lack of specific protocols for the evaluation of skeletal muscle structural and morphological properties in small animals has prompted researchers to adapt clinical and/or veterinary ultrasonographic devices for preclinical study on rodents. Thanks to huge technological advances, ultrasound systems specific for small animals are now available. In particular, the introduction of probes working at very high frequencies allowed for improvement in resolving power, making these new apparatuses suitable for preclinical studies on laboratory rodents. 

The possibility of investigating, on skeletal muscle, damage progression/changes in several conditions by using a non-invasive technique helps us to introduce new starting endpoints easily translatable to the clinical field. 

On one hand, ultrasound evaluations offering the possibility of performing longitudinal studies investigating skeletal muscle parameters on the same animal over time allow us to identify intra-individual differences as well as to reduce the number of animals necessary to obtain a statistically valid study. On the other hand, skeletal muscle ultrasound presents several limitations. As in the clinical field, preclinical ultrasound evaluations are also strongly related to operator skills, training, and experience. 

This, in addition to the lack of an independent validation of the technique, represents a major limitation towards the achievement of reference values for the main structural and functional parameters concerning skeletal muscle. 

With this review, we sought to gather information about procedures currently available in the literature on skeletal muscle preclinical evaluations by ultrasonography.

What emerged from our literature review was that the use of 40 MHz probes, characterized by a high resolution, is preferred to investigate structural parameters in rats such as volume, pennation angle, fiber length and muscle thickness. No data are yet available on the same parameters in mice. The 21 MHz probes, although reducing the resolution power, seem to be more adequate to accurately measure the 3D hind limb volume in mice, as an index of trophism, as well as the percentage of vascularization. In addition, the longitudinal position of the probe, with respect to the muscle to be analyzed, is one of the choices in all reported studies. Moreover, the best position of the animal to acquire images of posterior muscles of the hind limb (i.e., soleus or gastrocnemius) is the ventral decubitus, while the acquisition of hind limb anterior muscles (i.e., FDL or tibialis) is usually in the dorsal decubitus position. Despite this, the exact position of the hind limb is often not homogeneous in different studies, representing an important limitation to standardization of the procedure for evaluation of the structural readouts in rodent skeletal muscles. Importantly, for diaphragm muscle evaluations, a step toward protocol uniformity has been made already thanks to the independent validation of the technique carried out in two different laboratories [32,33].

## 5. Conclusions

By virtue of the extensive clinical use of ultrasonography for the evaluation of skeletal muscle readouts related to disease progression and therapeutic efficacy, the translational value of the application of this technique in preclinical settings is clear.

In this view, with our work, we want to encourage the independent optimization and validation of preclinical ultrasonography protocols on rodent skeletal muscle, particularly in hind limb. Moreover, similarly to what has been done with preclinical echocardiography, this could be a strong drive for the establishment of internationally validated standard operating procedures (SOPs) for rodent skeletal muscle ultrasonography in preclinical studies, similarly to the efforts made by the TREAT-NMD network (https://treat-nmd.org/). In fact, the need to create SOPs with a large independent consensus on the main methodological procedure and details on what the measurements will be and how they will be carried out, including a statistical analysis plan, is becoming more and more urgent. This could be useful in directing researchers towards more-targeted measurements, reducing the bias. Most importantly, the standardization of procedures and the repeatability of preclinical results would allow for significant improvements in clinical translatability and, in turn, clinical trials. Finally, in the coming years, a crucial milestone would be to make the use of preclinical ultrasound a gold standard technique for the acquisition and measurement of skeletal muscle properties.

## Figures and Tables

**Figure 1 ijms-24-04976-f001:**
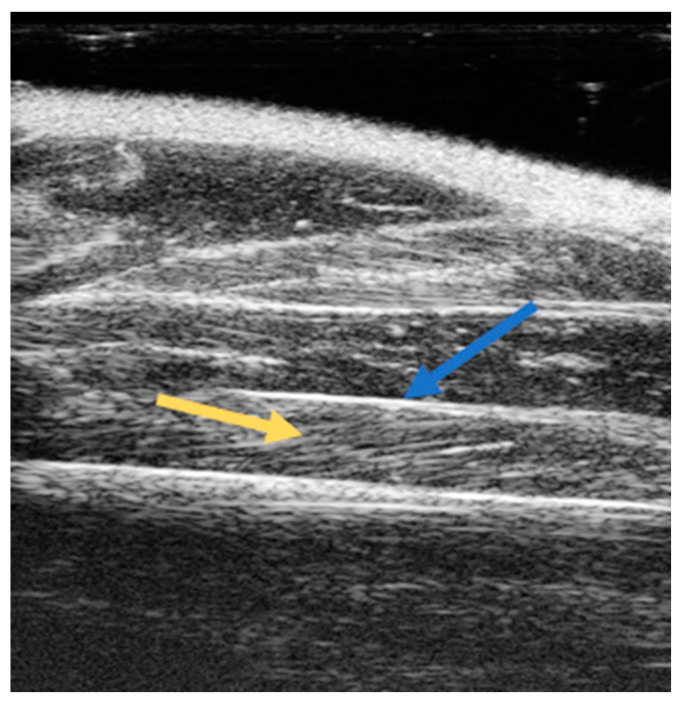
Sample image of soleus skeletal muscle longitudinal acquisition by ultrasonography. Image acquisition was performed in B-mode operating with a linear probe working at a frequency of 40 MHz. Lateral and axial resolutions were 80 and 40 µm, respectively. Soleus muscle is well distinguishable by the hyperechoic white structures of epimysium (blue arrow). The direction of muscle bundles is also clearly visible (yellow arrow).

**Figure 2 ijms-24-04976-f002:**
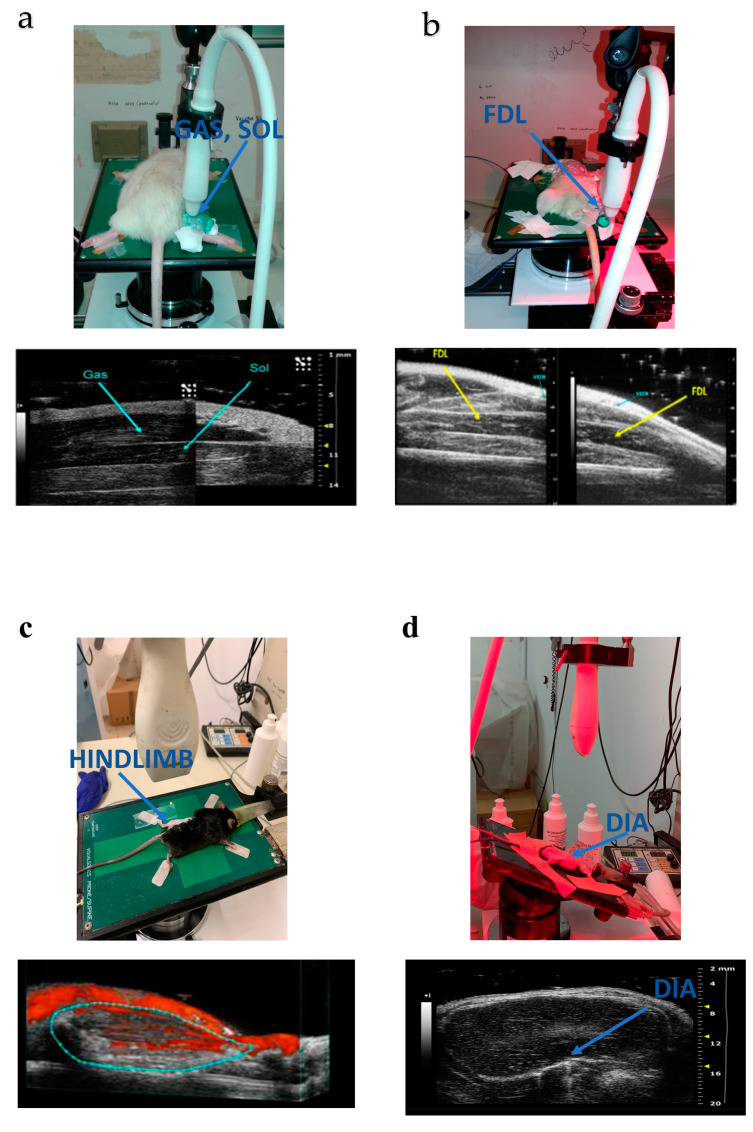
Main ultrasound views for skeletal muscle acquisition and evaluation of structural and functional parameters. (**a**) Rat placed in prone position (up) for GAS and SOL muscle acquisitions (down) used for volume PA, FL, and MT measurements. (**b**) Rat placed in supine position (up) for FDL muscle acquisitions (down) used for volume measurements. For both (**a**,**b**), image acquisition was performed in B-mode operating with a linear probe working at a frequency of 40 MHz. Lateral and axial resolutions were 80 and 40 µm, respectively. (**c**) Mouse placed in prone position (up) for 3-dimensional acquisitions of hind limb muscles in power Doppler mode (down) used for volume and percentage of vascularization measurements. Multiple 2-dimensional (2D) images were acquired at regular intervals in power Doppler mode by using an MS250 linear probe working at a frequency of 21 MHz, characterized by lateral and axial resolutions of 165 and 75 mm, respectively. At the end of the procedure, 3D images were reconstructed by Vevo2100 software 1.7.1. (**d**) Mouse placed in supine position (up) for DIA acquisitions (down) used for the evaluation of DIA movement amplitude and DIA echodensity. B-mode image acquisition was performed by using an MS250 linear probe working at a frequency of 21 MHz, characterized by lateral and axial resolutions of 165 and 75 mm, respectively.

**Figure 3 ijms-24-04976-f003:**
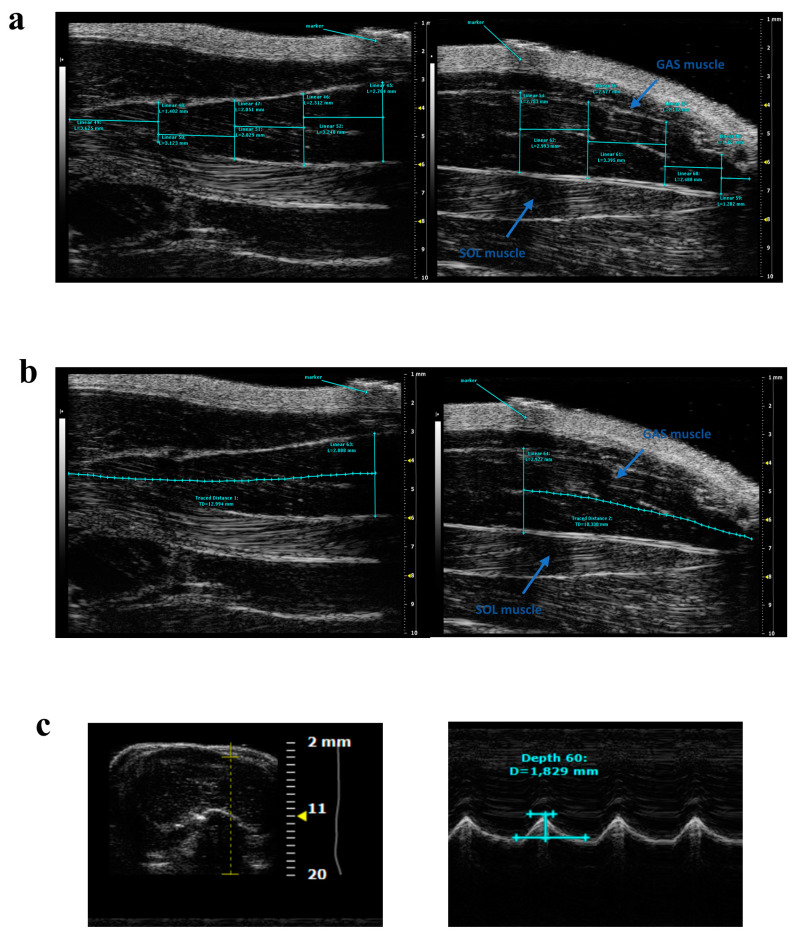
Representative ultrasonographic images of gastrocnemius, soleus, and diaphragm muscles and related measurements. (**a**) Proximal (left) and distal (right) acquisitions showing in cyan the measurements used to calculate the GAS volume by using the truncated cone method. (**b**) Proximal and distal acquisitions showing in cyan the measurements used to calculate the GAS volume by using the innovative sinusoidal method. (**c**) B-mode (left) and M-mode diaphragm acquisition. Diaphragm amplitude was measured in M-mode as the distance between the baseline and the peak of contraction. The B-mode images were used for echodensity evaluation. For methodological details, see Figure 2.

**Table 1 ijms-24-04976-t001:** Main features and uses of Vevo2100 probes.

	MS 250	MS 400	MS 550D
**Primary** **Application**	-Rat cardiology and abdominal (<400 g) -Large tumor imaging -All contrast applications	-Mouse cardiovascular -Rat abdominal -Rabbit eye -All vascular (mouse, rat, rabbit)	-Mouse cardiovascular -Mouse abdominal, reproductive -Mouse, rat embryology -Tumor imaging (up to 14 mm in diameter) -Small rat vascular, -Some abdominal (kidney)
**Skeletal** **Muscle** **Applications**	-Three-dimensional acquisition of mouse hind limb -Diaphragm function and morphology	-Diaphragm function	-Rat skeletal muscle structural parameters (PA, FL, MT) -Rat skeletal muscle volume
**Central** **Frequency**	21 MHz	30 MHz	40 MHz
**Image** **Width (max)**	23 mm	15.4 mm	14.1 mm
**Image** **Depth (max)**	30 mm	20 mm	15 mm
**Image** **Axial** **Resolution**	775 mm	50 mm	40 mm
**Image** **Lateral** **Resolution**	165 mm	110 mm	90 mm

**Table 2 ijms-24-04976-t002:** Summary of the main skeletal muscle applications of ultrasound.

Ultrasound Apparatus Probe Frequency	Muscle Type	Animal Position	Parameters	Ref.
Vevo 770 (VisualSonics, Toronto, ON, Canada) 40 MHz	GAS, SOL	Rat in ventral decubitus position with the ankle immobilized in full extension. Longitudinal acquisitions	PA, MT	[22]
Vevo 770 (VisualSonics, Toronto, ON, Canada) 40 MHz	GAS	Rat in ventral decubitus position with the ankle immobilized in an angle of 90°. Longitudinal acquisitions	PA, MT	[25]
LOGIQe; General Electric 8–14 Mhz	GAS	Rat in prone decubitus position Longitudinal and transversal acquisitions	Area of injured zone, fascial integrity, injured area borders	[26]
Vevo 770 (VisualSonics, Toronto, ON, Canada) 40 MHz	GAS	Rat in prone decubitus position Ankle at neutral flexion and the knee extended	Muscle swelling, thickness, and boundaries	[27]
SonoSite Titan Ultrasound (SonoSite Inc., Bothell, WA, USA) 5–10 MHz	GAS	Rat limb was placed in a steel pan and positioned at an angle of 20° to the pan side	MT, CSA	[28]
Philips iU22, NZE 737 (Philips Healthcare— Ultrasound, Eindhoven, the Netherlands) 7–15 MHz	GAS	Rat limb was placed in a steel pan and positioned at an angle of 20° to the pan side	MT, CSA	[28]
SonoSite Titan Ultrasound (SonoSite, Inc., Bothell, WA, USA) 5–10 MHz	Tibialis	Rat in dorsal decubitus by holding knee and ankle joints in a 90-degree angle, placing the probe in a 45-degree angle	CSA	[29]
Vevo 2100 (VisualSonics, Toronto, ON, Canada) 40 MHz	GAS, SOL	Rat in ventral decubitus position, with the hind limbs parallel to the body and with the foot forming an angle of 90° with respect to the hind limb, placing the probe parallel to the longitudinal muscle axis	Volume	[23]
Vevo 2100 (VisualSonics, Toronto, ON, Canada) 40 MHz	FDL	Rat in dorsal decubitus position, with the hind limb at its maximum extension and strictly parallel to the rat body, placing the probe parallel to the longitudinal muscle axis	Volume	[30]
Vevo 2100 (VisualSonics, Toronto, ON, Canada) 21 MHz	Hind limb	Mouse in ventral decubitus position, with the hind limbs parallel to the body and with the foot forming an angle of 90° with respect to the hind limb, placing the probe parallel to the longitudinal muscle axis	Volume and echodensity	[24,31]
Vevo 2100 (VisualSonics, Toronto, ON, Canada) 21–30 MHz	DIA	Mouse in dorsal decubitus position, with the platform tilted by 30° from horizontal with the head of the mouse lower with respect to the feet. The probe was fixed transversally to the mid sternal of the mouse at 120° with respect to the platform	Movement amplitude and echodensity	[32,33]

Abbreviations: GAS (Gastrocnemius Muscle); SOL (Soleus Muscle); PA (Pennation Angle); MT (Muscle Thickness); CSA (Cross Sectional Area); FDL (Flexor Digitorum Longus); DIA (Diaphragm).

## Data Availability

Not applicable.

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
