# Peer review of "Preclinical Ultrasonography in Rodent Models of Neuromuscular Disorders: The State of the Art for Diagnostic and Therapeutic Applications"

_ijms, 2023, doi:10.3390/ijms24054976_

Round 1

Reviewer 1 Report

1)  Abstract. L22-25. Despite this, only a few reports are available on skeletal mus-  cle ultrasound evaluations in small rodents. In this review, we describe the state of the art on this  technique’s skeletal muscle applications in preclinical studies, aiming to provide information for the  achievement of standard protocols and reference values useful in translational research on neuro-  muscular disorders. Please improve this paragraph.

2) 1. Background L31-36. Ultrasonography is a non-invasive and patient-friendly diagnostic imaging tech- nique that uses ultrasound. Thanks to its numerous advantages, namely use of non-ion- izing radiation, real-time display, portability and relatively low costs, ultrasonography is employed in clinical practice since 1950 in several fields of medicine, such as cardiology, gynecology, gastroenterology, oncology, etc. In 1980, Heckmatt et al. published a paper  in which the ultrasound technique was used for the first time to evaluate the structural changes of skeletal muscle. 

a-Recent Advances in Ultrasound Diagnosis of Carpal Tunnel Syndrome. Diagnostics 202010, 596. https://doi.org/10.3390/diagnostics10080596

b- Correlation between circulating fibrocytes and dermal thickness in limited cutaneous systemic sclerosis patients: a pilot study. Rheumatol Int. 2019;39(8):1369-1376. doi:10.1007/s00296-019-04315-7

c-Imaging of connective tissue diseases: Beyond visceral organ imaging?. Best Pract Res Clin Rheumatol. 2016;30(4):670-687. doi:10.1016/j.berh.2016.10.002

3) Introduction. L41-44. Indeed, skeletal muscle ultrasonography results very useful for the diag- nosis of several pathophysiological conditions allowing to monitor muscle damage and  changes, occurring in chronic neuromuscular disorders and/or related to physical activity and consequently to monitor efficiency of therapeutic interventions. Please, improve this paragraph and underline the aim of the study.

3) Figure 1. Sample image of skeletal muscle acquisition by ultrasound.  Each muscle is outlined by the hyperechoic white structures of epimysium (blu arrow). For each  muscle, the direction of muscle bundles is also clearly visible (yellow arrow).  Please, add some information on site of evaluation and the probe used such as MHz, linear/curve, gain.

4) Figure 2. Main ultrasound views for skeletal muscle acquisition and evaluation of structural and 443 functional parameters. a. Rat placed in prone position (up) for GAS and SOL muscles acquisitions 444 (down) used for volume PA, FL, and MT measurements. b. Rat placed in supine position (up) for 445 FDL muscle acquisitions (down) used for volume measurements. c. Mouse placed in prone position 446 (up) for the 3-dimensional acquisitions of hind limb muscles in power doppler mode (down) used 447 for volume and percentage of vascularization measurements. d. Mouse placed in supine position 448 (up) for DIA acquisitions (down) used for the evaluation of DIA movements amplitude and DIA 449 echodensity. Same comments of Figure 1.

5) Figure 3. Representative ultrasonographic images of gastrocnemius, soleus, and diaphragm mus- 452 cles. a. Proximal (left) and distal (right) acquisitions showing in cyan the measurements used to 453 calculate the GAS volume by using the truncated cone method. b. Proximal and distal acquisitions 454 showing in cyan the measurements used to calculate the GAS volume by using the innovative si- 455 nusoidal method. c. B-mode (left) and M-mode diaphragm acquisition. Diaphragm amplitude was 456 measured in M-mode as the distance between the baseline and the peak of contraction. The B-mode 457 images were used for echodensity evaluation. 458. Same comments of other figures.

6) Conclusions. L 487-493. In this context, a strong drive could come from the  standardization of preclinical studies. In fact, the need to create standard preclinical pro-  tocols including a statistical analysis plane, details on what the measurements will be and  how they will be carried out, is becoming more and more necessary. This could be useful  in directing researchers towards more targeted measurements easily translatable in clini-  cal by greatly reducing the bias. Finally, in the next years, a very important milestone  would be to make the ultrasound a gold standard technique for the acquisition and meas-  urement of skeletal muscle properties. Please underline the possible clinical implications.

Author Response

We thank the Referee for the careful revision of our manuscript and for the constructive observations that allowed us to improve it. Our responses on the raised issues, along with the main changes in the text, are listed below:

1)  Abstract. L22-25. Despite this, only a few reports are available on skeletal muscle ultrasound evaluations in small rodents. In this review, we describe the state of the art on this technique’s skeletal muscle applications in preclinical studies, aiming to provide information for the achievement of standard protocols and reference values useful in translational research on neuromuscular disorders. Please improve this paragraph.

ANS: We thank the Reviewer for this suggestion. We improved the paragraph accordingly.

2) 1. Background L31-36. Ultrasonography is a non-invasive and patient-friendly diagnostic imaging technique that uses ultrasound. Thanks to its numerous advantages, namely use of non-ionizing radiation, real-time display, portability and relatively low costs, ultrasonography is employed in clinical practice since 1950 in several fields of medicine, such as cardiology, gynecology, gastroenterology, oncology, etc. In 1980, Heckmatt et al. published a paper in which the ultrasound technique was used for the first time to evaluate the structural changes of skeletal muscle.

a-Recent Advances in Ultrasound Diagnosis of Carpal Tunnel Syndrome. Diagnostics 2020, 10, 596. https://doi.org/10.3390/diagnostics10080596.

b-Correlation between circulating fibrocytes and dermal thickness in limited cutaneous systemic sclerosis patients: a pilot study. Rheumatol Int. 2019;39(8):1369-1376. doi:10.1007/s00296-019-04315-7.

c-Imaging of connective tissue diseases: Beyond visceral organ imaging? Best Pract Res Clin Rheumatol. 2016;30(4):670-687. doi:10.1016/j.berh.2016.10.002.

ANS: We appreciate the Reviewer’s suggestions about the other applications of ultrasound. The reference list has been implemented.

3) Introduction. L41-44. Indeed, skeletal muscle ultrasonography results very useful for the diagnosis of several pathophysiological conditions allowing to monitor muscle damage and changes, occurring in chronic neuromuscular disorders and/or related to physical activity and consequently to monitor efficiency of therapeutic interventions. Please, improve this paragraph and underline the aim of the study.

ANS: We agree with the Reviewer that the general aim of the review needed to be better highlighted, and we improved the paragraph accordingly.

3) Figure 1. Sample image of skeletal muscle acquisition by ultrasound. Each muscle is outlined by the hyperechoic white structures of epimysium (blu arrow). For each muscle, the direction of muscle bundles is also clearly visible (yellow arrow).  Please, add some information on site of evaluation and the probe used such as MHz, linear/curve, gain.

ANS: As suggested, more information was added to Figure 1 caption.

4) Figure 2. Main ultrasound views for skeletal muscle acquisition and evaluation of structural and 443 functional parameters. a. Rat placed in prone position (up) for GAS and SOL muscles acquisitions 444 (down) used for volume PA, FL, and MT measurements. b. Rat placed in supine position (up) for 445 FDL muscle acquisitions (down) used for volume measurements. c. Mouse placed in prone position 446 (up) for the 3-dimensional acquisitions of hind limb muscles in power doppler mode (down) used 447 for volume and percentage of vascularization measurements. d. Mouse placed in supine position 448 (up) for DIA acquisitions (down) used for the evaluation of DIA movements amplitude and DIA 449 echodensity. Same comments of Figure 1.

ANS: As suggested, more information was added to Figure 2 caption. Further methodological details can be found in Table 2.

5) Figure 3. Representative ultrasonographic images of gastrocnemius, soleus, and diaphragm muscles. a. Proximal (left) and distal (right) acquisitions showing in cyan the measurements used to calculate the GAS volume by using the truncated cone method. b. Proximal and distal acquisitions showing in cyan the measurements used to calculate the GAS volume by using the innovative sinusoidal method. c. B-mode (left) and M-mode diaphragm acquisition. Diaphragm amplitude was measured in M-mode as the distance between the baseline and the peak of contraction. The B-mode images were used for echodensity evaluation. 458. Same comments of other figures.

ANS: As suggested, more information was added to Figure 3 caption. Further methodological details can be found in Table 2.

6) Conclusions. L 487-493. In this context, a strong drive could come from the standardization of preclinical studies. In fact, the need to create standard preclinical protocols including a statistical analysis plane, details on what the measurements will be and how they will be carried out, is becoming more and more necessary. This could be useful in directing researchers towards more targeted measurements easily translatable in clinical by greatly reducing the bias. Finally, in the next years, a very important milestone would be to make the ultrasound a gold standard technique for the acquisition and measurement of skeletal muscle properties. Please underline the possible clinical implications.

ANS: Thanks for this useful comment to improve the paragraph. Conclusions have been updated accordingly.

Reviewer 2 Report

The review article entitled "Preclinical Ultrasound in Rodent Models of Neuromuscular Disorders: The State of The Art for Diagnostic and Therapeutic Applications" aims to describe the state of the art in the application of the technique to skeletal muscle in preclinical studies and to provide information for the establishment of standard protocols and reference values useful for translational research on neuromuscular disorders. Because of these advantages, ultrasonography is widely used in sports medicine and neuromuscular diseases, such as myotonic dystrophy and Duchenne muscular dystrophy, to determine various structural and functional parameters of skeletal muscle. In line with this, I consider this review fit well scope of IJMS and state of art. The paper is well written and has the makings of a publication.

Minor: Include future strategies for establishing guidelines to promote better implementation in clinical trials.

Author Response

The review article entitled "Preclinical Ultrasound in Rodent Models of Neuromuscular Disorders: The State of The Art for Diagnostic and Therapeutic Applications" aims to describe the state of the art in the application of the technique to skeletal muscle in preclinical studies and to provide information for the establishment of standard protocols and reference values useful for translational research on neuromuscular disorders. Because of these advantages, ultrasonography is widely used in sports medicine and neuromuscular diseases, such as myotonic dystrophy and Duchenne muscular dystrophy, to determine various structural and functional parameters of skeletal muscle. In line with this, I consider this review fit well scope of IJMS and state of art. The paper is well written and has the makings of a publication.

Minor: Include future strategies for establishing guidelines to promote better implementation in clinical trials.

ANS: We thank the Reviewer for the positive comments on our manuscript and for the helpful suggestions to refine it. The text has been implemented accordingly.

Reviewer 3 Report

This manuscript presents a  review  related with the  describe the state of the art on this technique’s skeletal muscle applications in preclinical studies.

 This review  is generally well written, but it is unclear what this review adds to what is already known and have been published earlier. No clear research question seems to be formulated, the conclusions are unclear and other major concerns with this manuscript.

My specific comments are stated below. Overall, several important issues need to be addressed and some are of methodological character which requires a considerable revision of the paper. 

Abstract: The abstract should be a total of about 200 words maximum. The abstract should be a single paragraph and should follow the style of structured abstracts, but without headings: 1) Background: Place the question addressed in a broad context and highlight the purpose of the study; 2) Methods: Describe briefly the main methods or treatments applied. Include any relevant preregistration numbers, and species and strains of any animals used. 3) Results: Summarize the article's main findings; and 4) Conclusion: Indicate the main conclusions or interpretations. The abstract should be an objective representation of the article: it must not contain results which are not presented and substantiated in the main text and should not exaggerate the main conclusions.

Introduction

1. The introduction section did not provide a clear rationale for carrying out the study (for example, why is your research question important? What gap in the literature is the study addressing?

I suggest in this section in the first paragraph should have a sentence or two added that better outlines why this study is important.

In the last paragraph, the significance of the proposed word should be included highlighting why your work is important. what is the scientific contribution of this paper? it is not clear how this paper can make a significant contribution to the state of the art. 

In addition, author´s hypotheses should be included .

2. Methods: Was the study registrered at PROSPERO? Please to include the ID - number.

3.  Methods - literature search and selection: Please outline the exact search string or provide an appendix with the search strategy with specific search outcomes for each search and combinations. 

4. Methods - literature search and selection: Did you restrict study selection on any language? 

5. Results.Please add this section and to include info according to international standards, so they do  provide specific numerical data. Please add the calculate related with risk of bias was evaluated of this investigation.

6. Within your discussion,  compare outline your results, discuss their novelty and their application to practice.

7. Conclusion. These conclusions need to be softened, modified a in order to reflect only the study findings.

Author Response

This manuscript presents a review related with the describe the state of the art on this technique’s skeletal muscle applications in preclinical studies. This review is generally well written, but it is unclear what this review adds to what is already known and have been published earlier. No clear research question seems to be formulated, the conclusions are unclear and other major concerns with this manuscript.

My specific comments are stated below. Overall, several important issues need to be addressed and some are of methodological character which requires a considerable revision of the paper.

We thank the Reviewer for judging our review well-drafted. We understand the criticisms of the Reviewer about our manuscript. We to address the raised points in our specific answers as well as in the main text, whenever possible.

Abstract: The abstract should be a total of about 200 words maximum. The abstract should be a single paragraph and should follow the style of structured abstracts, but without headings: 1) Background: Place the question addressed in a broad context and highlight the purpose of the study; 2) Methods: Briefly describe the main methods or treatments applied. Include any relevant preregistration numbers, and species and strains of any animals used. 3) Results: Summarize the article's main findings; and 4) Conclusion: Indicate the main conclusions or interpretations. The abstract should be an objective representation of the article: it must not contain results which are not presented and substantiated in the main text and should not exaggerate the main conclusions.

ANS: The abstract has been structured according to the MDPI template here reported by the Reviewer, taking into account that it refers to a literature review and not to an original research article.

Introduction

  1. The introduction section did not provide a clear rationale for carrying out the study (for example, why is your research question important? What gap in the literature is the study addressing?

I suggest in this section in the first paragraph should have a sentence or two added that better outlines why this study is important.

In the last paragraph, the significance of the proposed word should be included highlighting why your work is important. what is the scientific contribution of this paper? it is not clear how this paper can make a significant contribution to the state of the art.

In addition, author´s hypotheses should be included.

ANS: We thank the Reviewer for pointing out these issues. The revised text has been implemented according to the Reviewer’s suggestions. We hope that these changes will help to highlight our work significance and objective. 

  1. Methods: Was the study registered at PROSPERO? Please to include the ID - number.
  2. Methods - literature search and selection: Please outline the exact search string or provide an appendix with the search strategy with specific search outcomes for each search and combinations.
  3. Methods - literature search and selection: Did you restrict study selection on any language?

ANS: As a literature review, our manuscript did not require the registration on the PROSPERO web site as for systematic reviews or meta-analysis. For the same reason, it did not contemplate a methods section. However, as normally done for literature reviews, we provided an overview of what is known on this particular topic. To do so, we used the PubMed database to screen literature results typing and crossing specific key words. Then, we selected papers in which ultrasonography was applied to the evaluation of structural and/or morphological skeletal muscle parameters with a well-described and reproducible methodology.

  1. Results. Please add this section and to include info according to international standards, so they do provide specific numerical data. Please add the calculate related with risk of bias was evaluated of this investigation.

ANS:  We thank the reviewer for the comments that further allow us to solve possible misunderstanding on the main aim of our review article. In fact, our review does not include the presentation of unpublished results. Results of cited literature, collected as previously described, are outlined in Table 2. Table 1 and Figures 1 – 3 are representative of the application of the ultrasound methodology described in the text.

  1. Within your discussion, compare outline your results, discuss their novelty and their application to practice.

ANS: As stated in the Introduction (background), the novelty of our review is that it represents the first report describing the state-of-the-art of studies in which ultrasound has been differently applied and validated for skeletal muscle measurements in mice and rats. Each literature result presented in the review has been discussed throughout the text.

  1. Conclusion. These conclusions need to be softened, modified in order to reflect only the study findings.

ANS: Taking into account the Reviewer’s suggestion, we remodulated the Conclusions.

Round 2

Reviewer 3 Report

I would like to thank the authors for their work, however I did not feel the authors made any significant improvements with regards to the main issues I raised in the first review. In its present state the paper provides no clear evidence that the authors propose a experiment must have been conducted rigorously. Also,  this research must been registered  with a record number in  PROSPERO is the International prospective register of systematic reviews or to indicate this is narrative review. In addition, the clinical relevance is not high and need to be given rationale.

The authors still need to make substantial changes inline with the issues raised in the first review. Indeed, the authors need add the sections of results and discussion should interpret the results including tables in line with the publications cited in the methods section and taking the limits of the study into account.

Author Response

I would like to thank the authors for their work, however I did not feel the authors made any significant improvements with regards to the main issues I raised in the first review. In its present state the paper provides no clear evidence that the authors propose a experiment must have been conducted rigorously. Also, this research must been registered with a record number in PROSPERO is the International prospective register of systematic reviews or to indicate this is narrative review. In addition, the clinical relevance is not high and need to be given rationale.

The authors still need to make substantial changes in line with the issues raised in the first review. Indeed, the authors need add the sections of results and discussion should interpret the results including tables in line with the publications cited in the methods section and taking the limits of the study into account.

We greatly thank the Reviewer for the precious time spent in reviewing our manuscript and providing insightful comments and valuable suggestions that allowed us to improve our work and better clarify its main goals. We understand the Reviewer’s criticisms and we tried our best to address each issue in the current version of the manuscript.

First we verified the possibility to register our work to the PROSPERO database and, in fact, we discovered that it does not respond to the inclusion criteria, since the database is exclusively dedicated to systematic reviews. For this reason, as suggested by the Reviewer, we better explained the nature of our work defining it as narrative (or literature) review throughout the text. Importantly, in the manuscript we wanted to underline that our review represents the first work aiming to provide the scientific community with the state-of-art about the most recent preclinical applications of ultrasonography to study skeletal muscle particularly in rodent models of neuromuscular disorders. This, with the hope that sharing these indications may help to promote a rigorous, independent validation of well-defined readouts, to ultimately accelerate translational research in this field, also considering that ultrasonography is a valid and reliable imaging technique for the assessment of muscle changes in various clinical settings.

We then tried to improve the manuscript along the useful criticisms raised by the Reviewer. In particular, to better organize the structure of the manuscript and clarify the approach used, we added a brand-new Methods section describing how literature search was conducted to construct the review article. Also, according to the Reviewer’s suggestion, all literature findings have been included in a dedicated Results section along with tables. These findings have been better interpreted by the authors both in the Discussion and Conclusions, also underlining the still existing limitations due to the lack of protocol uniformity, as well as the possible advantages deriving from a rigorous validation and international approval of the experimental procedures.

Round 3

Reviewer 3 Report

I am happy with the paper as it stands. Congratulations